# Methionine aminopeptidase 2 and its autoproteolysis product have different binding sites on the ribosome

Marius A. Klein[1], Klemens Wild ⓘ [1], Miglė Kišonaitė[1] & Irmgard Sinning ⓘ [1] ✉

Excision of the initiator methionine is among the first co-translational processes that occur at the ribosome. While this crucial step in protein maturation is executed by two types of methionine aminopeptidases in eukaryotes (MAP1 and MAP2), additional roles in disease and translational regulation have drawn more attention to MAP2. Here, we report several cryo-EM structures of human and fungal MAP2 at the 80S ribosome. Irrespective of nascent chains, MAP2 can occupy the tunnel exit. On nascent chain displaying ribosomes, the MAP2-80S interaction is highly dynamic and the MAP2-specific N-terminal extension engages in stabilizing interactions with the long rRNA expansion segment ES27L. Loss of this extension by autoproteolytic cleavage impedes interactions at the tunnel, while promoting MAP2 to enter the ribosomal A-site, where it engages with crucial functional centers of translation. These findings reveal that proteolytic remodeling of MAP2 severely affects ribosome binding, and set the stage for targeted functional studies.

In order to maintain the integrity of the cellular proteome, proper protein modification, localization, and folding must be ensured. Despite the fast speed of translation, cells manage to carry out these essential functions while protein biosynthesis is still ongoing, thus coupling mRNA translation and protein maturation[1]. To elucidate the ordered sequence of events that matures an unstructured nascent chain into a fully functional protein, many studies have focused on the structural characterization of ribosome-associated factors (RAFs)[1]. The signal recognition particle (SRP) has been among the first RAFs studied in detail in context of the ribosome[2,3], and recently in combination with the nascent polypeptide associated complex (NAC), which revealed their interplay at the peptide tunnel exit (PTE)[4]. Structures of the ribosome-associated complex (RAC)[5,6] provided insights into chaperone binding and dynamics at the 80S ribosome. However, our understanding of early co-translational processing at the eukaryotic ribosome is lagging behind. The cryo-EM structure of the N-terminal acetyltransferase A (NatA) complex from *S. cerevisiae* gave insights into enzyme interactions at the PTE[7]. NatA does not function independently and requires N-terminal methionine excision (NME) for activity. How the essential mechanism of NME is carried out by methionine aminopeptidases (MAPs) at the PTE is largely unexplored

MAPs are among the first RAFs to co-translationally interact with the nascent chain as it emerges from the PTE. These metalloproteases catalyze the proteolytic removal of the initiator methionine[8], which constitutes an important protein turnover signal[9]. Unlike bacteria and archaea, eukaryotes express two types of MAPs[10], with highly similar substrate specificities[11]. While MAP1 and MAP2 share the common pita-bread fold, the latter is additionally characterized by the presence of several insertions and a charged N-terminal extension, which consists of segmented regions of alternating poly-acidic and poly-basic stretches (Fig. 1a)[12]. This N-terminal extension of MAP2 shows substantial length and sequence variations between different organisms[13], and its precise function is only poorly understood. Furthermore, MAP2 recruitment and activity was shown to be influenced by the rRNA expansion segment ES27L[14].

Aside from its activity as a ribosome-associated protease, MAP2 was initially identified as a eukaryotic initiation factor 2 (eIF2) associated glycoprotein, with regulatory functions related to translational initiation and cellular growth[15]. Through elaborate studies, these additional functions could be assigned to the glycosylated N-terminal extension, while its proteolytic activity resides in the conserved core domain[16]. A reported autoproteolytic activity of MAP2, that can

[1]Heidelberg University Biochemistry Center (BZH), Im Neuenheimer Feld 328, 69120 Heidelberg, Germany. ✉e-mail: irmi.sinning@bzh.uni-heidelberg.de

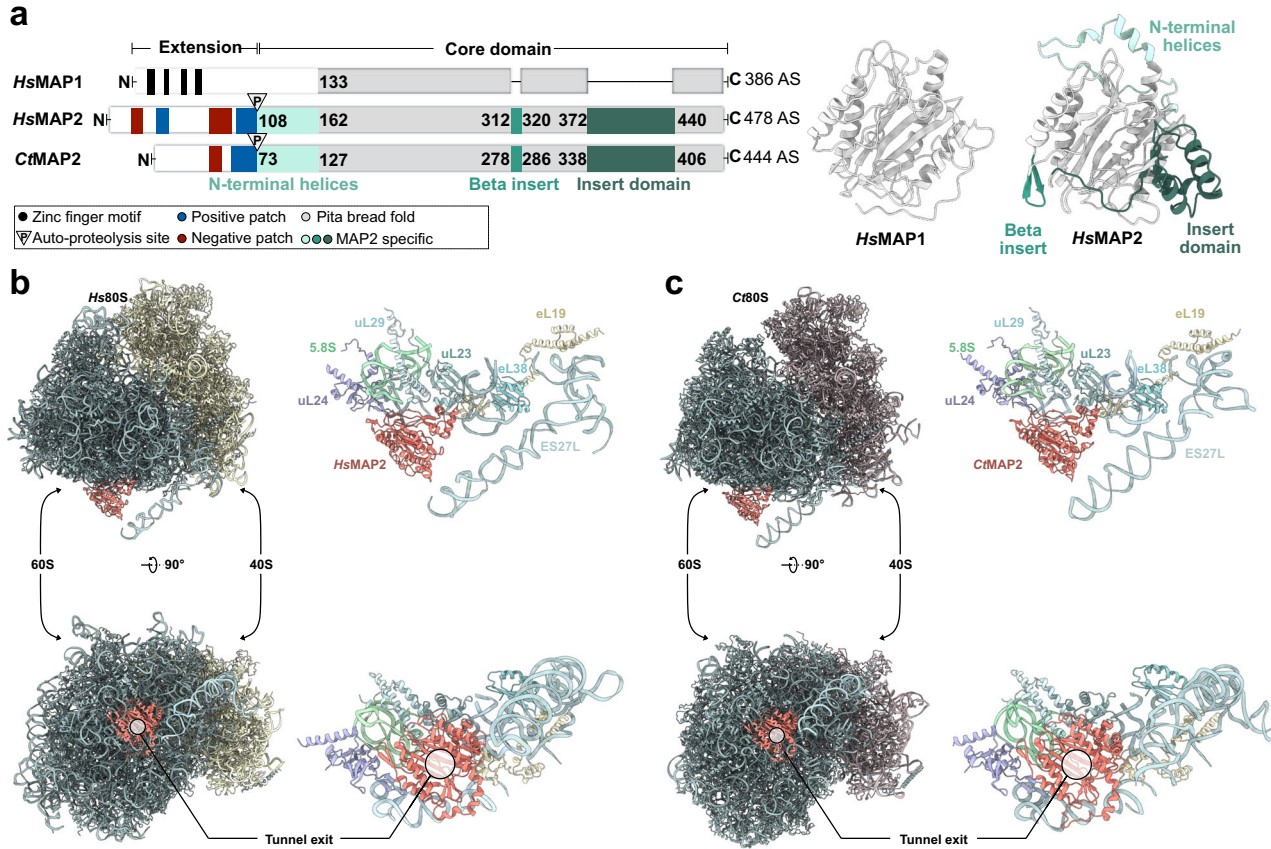

**Fig. 1 | Cryo-EM structures of MAP2 at the ribosomal tunnel exit. a** Domain organization of *Hs*MAP1, compared to *Hs*MAP2 and *Ct*MAP2. Alphafold predictions of *Hs*MAP1 and *Hs*MAP2 are shown. The plain pita-bread fold is shown in gray. Structural features and insertions unique to MAP2 are highlighted in shades of green. Side- and top-view of (**b**) *Hs*MAP2 and (**c**) *Ct*MAP2 (shown in red) bound to

the PTE and ES27L. The overviews on the left show the large 60S and small 40S subunit as well as the position of the tunnel exit. Tunnel exit proteins uL24, uL29, uL23, eL19, eL38 and parts of the 28S and 5.8S rRNA were built and are shown on the right side of the panels.

physically separate these functions, might add an additional layer of complexity onto the functional regulation of this protein[16].

When upregulated, MAP2 stimulates cancer cell proliferation and seems to play an important role in tumor progression[17]. The discovery of MAP2 as a druggable target of angiogenesis spurred decades of research into selective inhibitors and yielded several high-resolution X-ray structures in complex with promising candidate compounds (for review see ref. 18). However, none of the MAP2 structures have been solved in the ribosomal context.

Previous cryo-EM structures of catalytically inactive MAP2-like proteins Arx1[19] and Ebp1[20,21] bound to 80S ribosomes revealed the importance of a MAP2-specific insert domain as main mediator of ribosome binding. Arx1 and Ebp1 share their binding site on top of the tunnel exit and recruit the long eukaryotic rRNA expansion segment ES27L. Since many ribosome-associated factors use this binding site and also recruit ES27L[7,19,21], dynamic mechanisms are likely required to enable an exchange of factors during protein biosynthesis at the ribosome. How the handover of the nascent chain, the rearrangement of these factors, and the interplay with ES27L is coordinated at the tunnel exit, is not understood. Recently, a cryo-EM structure of mammalian MAP1 at the PTE revealed a dependence on NAC, which was proposed to recognize the zinc-finger motifs at the N-terminus of MAP1 with the C-terminus of the NAC-beta subunit[22]. In contrast, such a factor dependence was not found in yeast, as MAP1 was captured at a different binding site without NAC[23]. The molecular details of MAP2 binding, and the differences found from higher to lower eukaryotes have remained elusive.

In order to shed light on this intricate system, we started to investigate how eukaryotic MAP2 binds the ribosomal tunnel exit. By concurrently analyzing this interaction for two eukaryotic organisms, we aimed for a structural comparison and the analysis of conserved features required for ribosome binding. In addition, we set out to investigate, what consequences might arise from the proposed autoproteolytic activity of MAP2[16] with respect to its interaction with the 80S ribosome. We determined high-resolution cryo-EM structures of MAP2 from *H. sapiens (Hs)* and the thermophilic fungus *C. thermophilum (Ct)* bound at the tunnel exit of 80S ribosomes. While *Hs*MAP2 binds in a single, rigid conformation on translationally inactive ribosomes, *Ct*MAP2 binding is highly dynamic in the presence of random nascent chains. We find that after autoproteolytic cleavage, the truncated *Ct*MAP2 variant can no longer recruit ES27L while binding to the PTE is impaired. Surprisingly, removal of the poly-charged N-terminal region enables *Ct*MAP2 to occupy the A-site in the non-rotated state, an interaction that is mediated by conserved structural features that are unique to MAP2.

## Results

### Structural basis of MAP2 binding to 80S ribosomes

To obtain a high-resolution structure and unveil the molecular details of MAP2 recruitment to the ribosome, we followed a combined approach of single particle cryo-EM, X-ray crystallography and the modeling tool Alphafold[24]. For cryo-EM of MAP2-containing complexes, human and *C. thermophilum* 80S ribosomes were purified and mixed with a large access of recombinant protein. Cryo-EM

reconstructions have been refined to an average resolution of 2.92 Å and 2.94 Å for the *H. sapiens* and *C. thermophilum* samples, respectively (Supplementary Figs. 1-4, and Supplementary Table 1). In the human complex, *Hs*MAP2 binds in a rigid manner and could be well resolved at a local resolution of ~3.5 Å after local refinement. In contrast, *Ct*MAP2 binding exhibited large conformational heterogeneity to the point where only parts of the insert domain could be resolved upon 3D-refinement. By performing focused 3D variability analysis[25], the large dataset could be separated into more homogenous subsets of particles to obtain well-defined 3D refinements. In combination with local refinements around ES27L and *Ct*MAP2, the resolution of both maps could be substantially increased. In order to build *Ct*MAP2 into the cryo-EM density, the crystal structure of *Ct*MAP2(ΔN2-73) was solved at 1.3 Å resolution (Supplementary Fig. 5 and Supplementary Table 2). Finally, Alphafold models were generated to predict the structures of the poly-charged N-terminal regions for *Hs* and *Ct*MAP2. In combination, these tools allowed for a detailed structural analysis of MAP2 bound to the ribosome.

*Hs*MAP2 and *Ct*MAP2 locate centrally on top of the tunnel exit, exposing their concave surfaces including the active sites to form a vestibule-like structure (Fig. 1b, c and Supplementary Fig. 6). The central placement allows for a minimal distance of the active site to nascent chain substrates (length <40 residues) when emerging from the tunnel. The MAP2 termini, as far as resolved, are pointing away from the ribosomal surface. Generally, the vestibule is almost closed leaving a substantial lateral gap only towards helices H24 and H47 (Supplementary Fig. 6). Furthermore, MAP2 is sandwiched between the tunnel exit and ES27L, which folds on top of the protein. In both MAP2/80S cryo-EM structures, the positively charged N-termini could not be resolved due to their inherent flexibility. When creating Alphafold models of full-length MAP2 proteins and subsequent superposition on the MAP2 structures of the cryo-EM reconstructions, the poly-lysine stretches close to the core domain are found to contact

ES27L. For *Ct*MAP2, the trajectory of the N-terminus passes perfectly through a widened major grove of ES27L (Supplementary Fig. 7), which was well defined in the local refinement. However, for the *Hs*MAP2 structure, ES27L displayed higher flexibility and the interaction mode with the *Hs*MAP2 N-terminus remains vague.

## MAP2 contacts at the ribosomal tunnel exit

As high-resolution structures are now available for the individual binding partners and the respective *Hs* and *Ct*MAP2/80S complexes, the determinants and dynamics of recruitment and binding can be examined in detail. Conserved MAP2 interaction partners around the tunnel exit include proteins eL19, uL23, uL24, uL29, 5.8S rRNA, and 28S rRNA (helices H24, H53, and H59) (Supplementary Fig. 6). The catalytic core domain of MAP2 with the typical pita-bread fold is mainly involved in charged interactions with the backbone of 28S rRNA and 5.8S rRNA, as well as with residues of uL24 (Fig. 2a and Supplementary Fig. 6). However, the major determinant for binding to the tunnel exit is the MAP2-specific insert domain of ~60 residues, which is locked in between uL23, the universal binding site for nascent chain associated factors, eL19, and 28S rRNA helices H53 and H59 (also termed ES24L) (Fig. 2b and Supplementary Fig. 6). The resulting interface of ~1000 Å² comprises 2/3ʳᵈ of the contacts to the tunnel exit. Although the rim of the tunnel exit is relatively even, especially H59 is protruding, thus creating a specific docking site for the insert domain. Most prominent, a conserved arginine at the apex of the insert (*Hs*MAP2 Arg395, *Ct*MAP2 Arg361) stacks on top of the closing tetraloop of H53 (*H. sapiens* C2526, *C. thermophilum* A1582) in between H59, eL19 and uL23 (Fig. 2b, c and Supplementary Fig. 8). This residue is however not conserved in other MAP2-like proteins like human Ebp1 or yeast Arx1, which carry a lysine in this position (Supplementary Fig. 9). Furthermore, the MAP2-uL23 contact of the insert domain reveals a hydrophobic core (Fig. 2d and Supplementary Fig. 6). The MAP2 interactions with eL19 and H59 are substantial, but apparently not conserved from

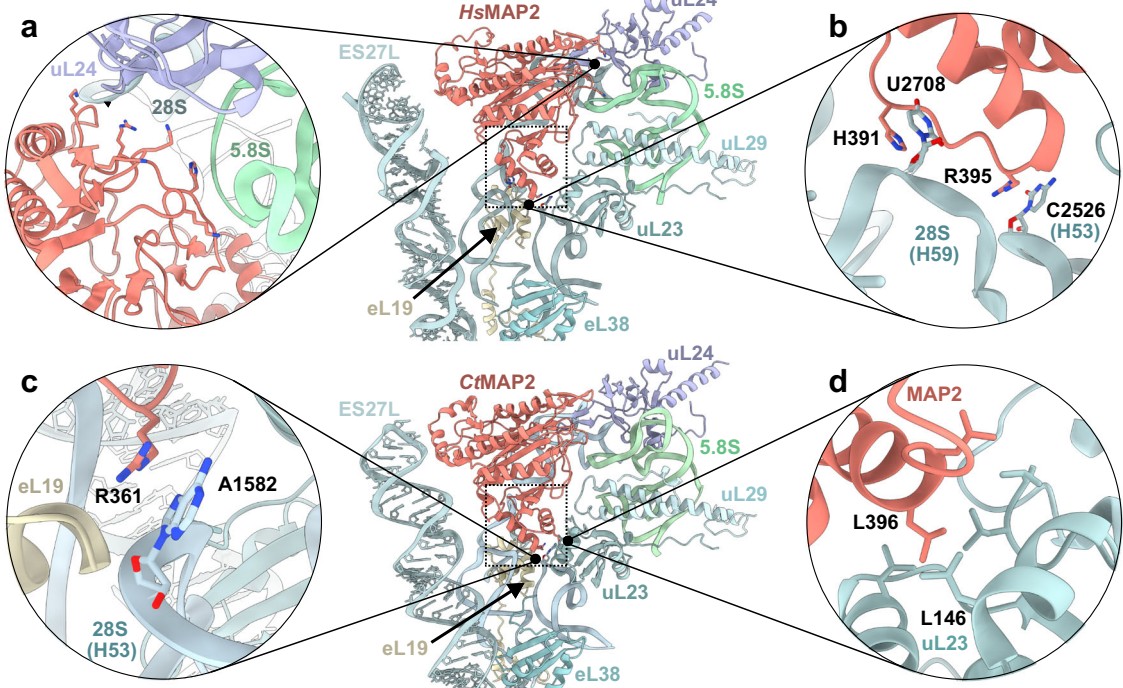

**Fig. 2 | MAP2 interactions at the ribosomal tunnel exit.** The central panels show an overview of *Hs*MAP2 and *Ct*MAP2 binding to the PTE. MAP2 is shown in red and the insert domain is indicated by the dashed box. **a** The core domain of *Hs*MAP2 electrostatically interacts with the 28S and 5.8S rRNA through several charged residues. **b** The insert domain of *Hs*MAP2 is fixed on 28S rRNA H53 (R395 stacks onto C2526) and binds specifically to H59 at U2708. Unlike *Ct*MAP2, *Hs*MAP2 tightly interacts with H59, which undergoes large conformational changes upon MAP2 binding. **c** Main contact points between *Ct*MAP2 and the ribosome include interactions of R361 with the 28S rRNA. **d** Hydrophobic interactions involving leucine residues between tunnel exit protein uL23 and *Ct*MAP2.

lower and higher eukaryotes. In the human case, H59 undergoes a large conformational remodeling with the closing loop forming multiple interactions with the insert domain (Supplementary Fig. 10). Herein, U2708 forms the main and sole specific contact by piercing into the insert domain where it stacks in between His391 and Leu403 (Fig. 2b and Supplementary Fig. 11). Similar H59 remodeling and interactions have also been described for the Ebp1/80S complex[21] (Supplementary Fig. 12). Interestingly, the binding of *Ct*MAP2 does not trigger a remodeling of H59, which remains in the same conformation as in the idle 80S structure (Supplementary Fig. 10). This observation goes in-line with the rigidity of H59 in *S. cerevisiae* in presence of Arx1[19] (Supplementary Fig. 12). Notably, the other two MAP2-specific insertions do not actively contribute to PTE binding (Supplementary Fig. 6).

### Dynamic rearrangements of MAP2 and ES27L

MAP2 recruitment and binding is part of the dynamic process of protein biosynthesis and maturation. Continuous elongation and subsequent binding of other RAFs necessitates rapid dynamic rearrangements of MAP2 binding, which can be deduced from our cryo-EM reconstructions. *Hs*MAP2 binding to the idle ribosome is rigid in the absence of an emerging nascent chain. In contrast, *Ct*MAP2 binding displays a large degree of variability resulting in a noisy map insufficient for building an accurate atomic model. Of note, *Ct*80S ribosomes treated with puromycin as used for the cryo-EM reconstruction do not release the nascent chain[26] (Supplementary Fig. 13) and the emerging nascent chains likely contribute to the mobility of *Ct*MAP2. Only parts of the insert domain closest to the apical Arg361 were well resolved by the overall refinement, indicating that this residue serves as a stable pivot point around which MAP2 can rotate at the tunnel exit (Fig. 3a). Indeed, by superimposing rigid body fitted MAP2 in the most extreme states obtained from focused 3D variability analysis, Arg361 appears as an invariant point of MAP2 rotation.

The available binding-space of MAP2 is limited by steric constraints (Fig. 3b, c). When rotated towards the ribosome, *Ct*MAP2 appears best resolved in the local refinement. The rotation is limited by a contact formed by His316, which touches down onto the phosphate backbone of rRNA H24 on the other side of the tunnel exit in respect to Arg361 (Fig. 3b and Supplementary Fig. 14a, b). Measuring the angle of rotation between Arg361 and His316 in the two extreme rotational states of MAP2, a maximum of 15° rotation was determined, which is due to constraints given by the sliding of the MAP2 insert domain over the uL23 surface. Here, MAP2 Leu396 translates by 5 Å towards uL23 Leu146 (Fig. 3c and Supplementary Fig. 14c, d), and a more extensive displacement would result in a steric clash. The second constraint is formed by the protruding 28S rRNA H59. As the rotation is towards this helix, it forms a barrier and MAP2 might finally be sheared-off (Supplementary Fig. 14e).

In our reconstruction, the highly flexible ES27L was best resolved in the conformation closest to the tunnel exit, where it is stabilized by interactions with the MAP2 N-terminal extension, and at an internal RNA bulge by contacts with protein eL38 (Fig. 3a). Accordingly, both components become destabilized as MAP2 rotates away from the tunnel exit. In this regard, the interaction seems to differ from the MAP2-like protein Ebp1, which forms more rigid contacts to ES27L[21] (Supplementary Fig. 12).

### MAP2 autoproteolysis product has a second binding site on the ribosome

Having studied the interactions of MAP2 at the ribosomal tunnel exit for two eukaryotic organisms, we were intrigued to investigate how the physiologically relevant autoproteolysis of MAP2, which removes the N-terminal extension, would affect this interaction. To confirm that *Ct*MAP2 and *Hs*MAP2 also undergo autoproteolysis, as previously shown for rat MAP2[16], purified protein was incubated at 4 °C and analyzed by subsequent SDS-PAGE and Mass Spectrometry. Both

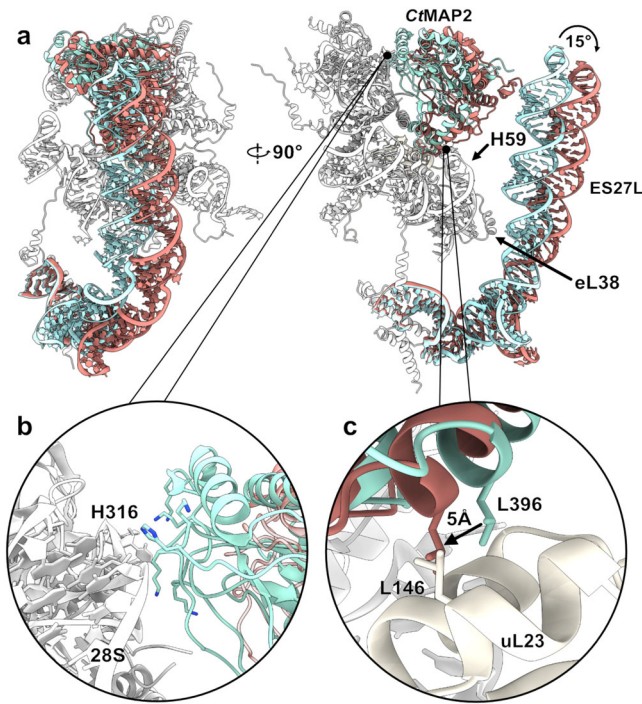

**Fig. 3 | Variability of *Ct*MAP2 and ES27L at the ribosomal tunnel exit. a** Front and side view of the MAP2/PTE interaction in the two most extreme states of its 15° rotation. Ribosomal proteins and rRNA are shown in shades of gray, while MAP2 and ES27L are colored in red and cyan for the two extreme states. Insets show specific interactions upon MAP2 rotation. **b** When fully rotated towards the ribosome, electrostatic interactions with the 28S rRNA stabilize *Ct*MAP2 at the tunnel exit around H316. **c** The rotation of *Ct*MAP2 at the tunnel exit brings uL23 L146 in close proximity to MAP2 L396, closing a lateral gap of ~5 Å.

*Ct*MAP2 and *Hs*MAP2 showed a complete degradation of the N-terminus up to the core domain, consistent with the previously reported autoproteolysis of rat MAP2 (Supplementary Figs. 15–17). Interestingly, we find that *Hs*MAP2 first degrades into two stable fragments, and eventually completely loses the N-terminal extension. Our cryo-EM structure of *Ct*MAP2 suggests a stabilizing function of ES27L during the rotation of MAP2. Without its charged N-terminal extension, the MAP2 core domain should not be able to recruit ES27L. To further examine the role of ES27L in MAP2 binding, a cryo-EM structure of the corresponding *Ct*MAP2(ΔN1-73; referred to as ΔN) construct bound to the ribosome was determined, following the same workflow as applied for the full-length protein. In this dataset, only about 70% of the 80S ribosomes contain *Ct*MAP2ΔN at the PTE (Fig. 4 and Supplementary Figs. 18 and 19). Notably, the overall binding mode of MAP2ΔN did not drastically change in this position, and pivoting around the insert domain could also be observed, as described for full-length MAP2. The most prominent difference at the PTE was the absence of the ES27L interaction. Without its N-terminal extension, MAP2ΔN can no longer engage in stabilizing interactions with ES27L, causing it to be more dynamic and consequently invisible in the final refinement.

However, after separation of the particles according to their translocation states by heterogeneous refinement, we discovered a second binding site for *Ct*MAP2ΔN. While in the rotated ribosomal state, the A-site was found occupied by the translocation factor eEF2 (Supplementary Fig. 18), in the non-rotated state *Ct*MAP2ΔN had entered this position (Fig. 4 and Supplementary Figs. 18 and 20). Unlike the highly adaptive and dynamic binding mode of MAP2 at the PTE, *Ct*MAP2ΔN is strongly stabilized at the A-site, resulting in a local resolution of ~3.4 Å. Towards the 40S subunit, mostly polar and charged interactions are formed by the rim of the concave side of

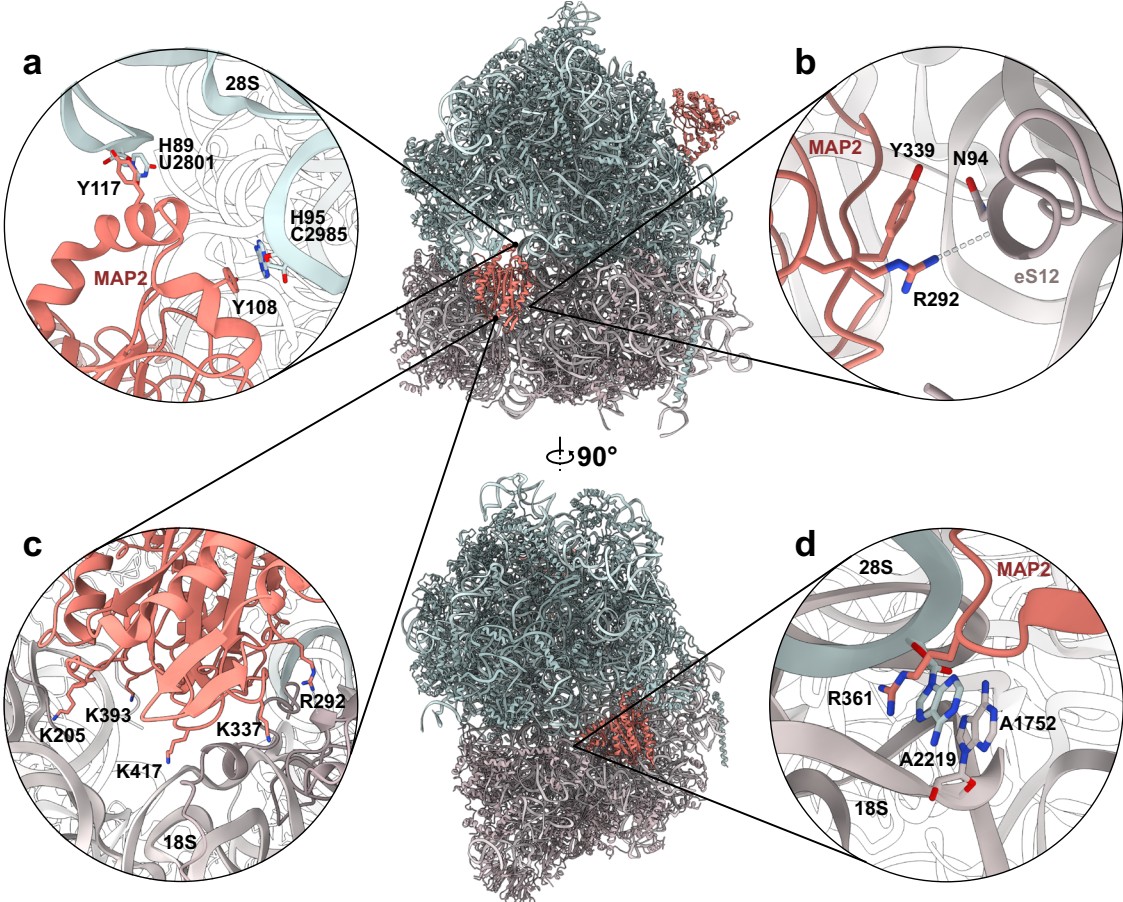

**Fig. 4 | CtMAP2ΔN can enter the A-site of Ct80S ribosomes.** Central panels show a front and side view onto the A-site interaction of CtMAP2ΔN. The top panel shows a second copy of CtMAP2ΔN at the canonical binding site at the PTE, but ES27L is not resolved. CtMAP2ΔN is shown in red, while the 60S and 40S subunit are colored in blue and gray, respectively. Insets show details of A-site binding. **a** Two tyrosine residues Y108 and Y117 form major contacts to the 28S rRNA through π-stacking interactions. **b** Contacts with 40S subunit protein eS12. **c** Ionic interactions with the charged phosphate backbone of the 18S rRNA. **d** The conserved arginine R361 penetrates deeply into the A-site and stacks onto the decoding center of the ribosome, where it induces conformational changes in A2219.

CtMAP2ΔN to protein eS30 and the extended N-terminus of eS23 (Supplementary Fig. 20), which becomes ordered in between the decoding center and the MAP2 insert domain. Towards the 60S subunit, two exposed tyrosines (Tyr108 and Tyr117) located in the N-terminal helices of CtMAP2ΔN contact the closing loops of helices H89 and H95 of 28S rRNA (Fig. 4a and Supplementary Figs. 21 and 22). These π-stacking interactions stabilize the flexible N-terminal helices of CtMAP2 (104-122), which are often not fully resolved in MAP2 X-ray structures. Strikingly, these residues, which form the major contact to the 60S subunit in the A-site position, serve no active role in MAP2 binding at its canonical binding site at the PTE (Supplementary Fig. 6). The N-terminal helices are only conserved amongst the catalytically active MAP2s and not for Ebp1 or Arx1. Of note, H95 corresponds to the sarcin-ricin-loop (SRL), the most highly conserved part of 28S rRNA involved in binding and stimulation of translation factor GTPases.

The most prominent protein-protein contact of CtMAP2ΔN formed in the A-site occurs with 40S protein eS12 (Fig. 4b), while additional electrostatic interactions are present via positively charged residues of the MAP2 core and the 18S rRNA (Fig. 4c). Alike for the PTE interaction, no contacts are formed between the beta insert of CtMAP2ΔN and the A-site, indicating that this conserved region has a different, yet unresolved function. In the A-site site, the negatively charged surface of CtMAP2ΔN is solvent exposed (Supplementary Fig. 21) and might electrostatically affect binding of other potential A-site interactors.

Notably, the conserved Arg361 that plays a major role in tunnel exit binding, now penetrates deepest into the A-site and interacts with the decoding center of the ribosome by stacking on two conserved adenines (A1752, A2219) (Fig. 4d). Adenine A1752 within 18S rRNA helix H44 is known to canonically read out the codon-anticodon helix[27] and is also bulged out in the absence of MAP2[26]. In contrast, adenine A2219 within the 28S rRNA is conformationally remodeled and solvent exposed only upon MAP2 binding.

The N-terminus of CtMAP2ΔN points towards the P-stalk of the 60S subunit and is exposed to the solvent. Therefore, it is not obvious, why the full-length protein would not be capable of A-site binding. When a respective Alphafold model including the N-terminus is superimposed, the flexible extension is placed into the space between the 40S and 60S subunit. However, as the MAP2 N-terminal extension is highly charged and also comprises stretches with negative charges, a repulsion especially from rRNA structures might restrict binding of full-length MAP2. More detailed analysis of the full length CtMAP2 dataset revealed that ~5% of 80S ribosomes also contained MAP2 in the A-site. In comparison, the percentage of CtMAP2ΔN in the A-site was four-fold higher. From these data it is not clear, whether full length MAP2 enters the A-site with lower affinity, or whether the ~5% of MAP2 found at the A-site were a result of the proteolytic removal of the N-terminal extension (Supplementary Fig. 3).

## Discussion
### MAPs and the ribosomal tunnel exit
All MAPs and MAP-like proteins share the pita-bread fold of the catalytic core domain, while the poly-charged N-terminal extension and

several insertions in the core domain are only present in MAP2. In the MAP2/80S structures presented here, two of these MAP2-specific features - the N-terminal extension and the insert domain - are identified as main ribosome interactors and position the catalytic core domain centrally on top of the PTE, while the remaining MAP2-specific features, the N-terminal helices and the beta insert, do not engage in interactions at this binding site. In contrast to MAP2, the plain pita-bread fold of *E. coli* MAP1 does not exhibit a robust association with the ribosome[28]. Consistently, our attempts to determine a cryo-EM structure of MAP1 bound to either *C. thermophilum* or *H. sapiens* ribosomes remained unsuccessful. This indicates that *Ct*MAP1 and *Hs*MAP1 seem to either require a specific substrate or other factors that assist in its recruitment and stable association with the PTE, while its binding mode likely differs from MAP2. For yeast MAP1, binding to the ribosome was shown to be undisturbed by MAP2, suggesting a different binding mode between the two proteins[29]. While this manuscript was under review, these hypotheses were validated by two cryo-EM studies of MAP1 bound to programmed 80S RNCs, with and without NAC[22,23]. On mammalian ribosomes, NAC was found to serve as HUB controlling MAP1 recruitment. In both structures, MAP1 sits off-center from the PTE and forms a less rigid interaction with the ribosome (Supplementary Fig. 23). Accordingly, the insert domain appears to bestow the ability of MAP2 and MAP2-like proteins to closely bind to the PTE both factor and substrate independent. These observations raise the questions on how MAP1 and MAP2, despite having a similar catalytic activity in NME, differ in their requirements for, and details of ribosome association as well as in the regulation of ribosome binding.

## The insert domain enables adaptive binding to the ribosome

Many factors share the PTE as a binding site[1] and need to quickly assess if the emerging nascent chain qualifies as a substrate that can be subjected to modification, folding or targeting. With translation progressing rapidly, a highly-coordinated exchange is required to ensure proper maturation of the nascent chain and to prevent ribosomal stalling or translation errors. In the absence of a nascent chain, as observed for our human MAP2/80S complex, MAP2 adopts a rather rigid binding mode. Since NME by MAPs is among the first co-translational processes that occur when the nascent chain emerges at the ribosome, it seems plausible that MAP2 could position itself in a "scanning mode" even before a substrate is displayed. Additionally, early occupation of the PTE by MAP2 might prevent a premature association of other factors such as SRP, NAC, RAC, or NATs. However, the interplay between these factors still needs to be addressed.

In the presence of nascent chains, as displayed by the *Ct*80S ribosomes, MAP2 adopts a dynamic binding mode by rotating away from the PTE. The pivoting of *Ct*MAP2 occurs around Arg361 at the apical end of the insert domain. Rotation with a similar amplitude and direction around the insert domain has been characterized also for Ebp1 by comparing multiple Ebp1-ribosome structures[20,21], and therefore seems to be a general feature of MAP2-like folds bound to the PTE. In order to understand the complete mechanisms that modulate MAP2-like protein binding, the driving forces for the rotation need to be clarified.

All MAP2-like proteins analyzed to date contact the long rRNA expansion segment ES27L with their convex outer surface. ES27L is intrinsically flexible and can reach up to the PTE to recruit and accommodate various factors to assist in protein biosynthesis. These protein-RNA interactions rely on the charged protein termini often containing arginine- or lysine-rich motifs. For *Ct*MAP2 the modeled poly-lysine stretch of this extension appears to pass through the RNA major groove next to the closing tetraloop of ES27L. To address the question, whether the rotation of *Ct*MAP2 is initiated by a force exerted by the recruited ES27L, we generated the physiologically relevant autoproteolysis product, which lacks the N-terminal extension. In the resulting cryo-EM structure, the binding mode at the tunnel exit was

virtually unchanged and MAP2 still performed a substantial rotation around the insert domain. These findings strongly indicate that the inherent variability of MAP2 is not caused by a pulling force exerted by ES27L. However, without the ES27L-recruiting N-terminal extension, binding is attenuated, reflected in the lower occupancy of this binding site. Another possible explanation for dynamic MAP2 interaction is the presence of random nascent chains as displayed by the *Chaetomium* ribosomes. When the nascent chain proceeds to grow beyond the distance to the active site, MAP2 might experience a pushing force that initiates the backward rotation around its insert domain. As MAP2 pivots, interactions at the tunnel exit are weakened and stabilization by ES27L might become increasingly relevant. Furthermore, species-specific differences in the architecture of the *Chaetomium* PTE (such as 28S rRNA H59) might also contribute to the dynamic MAP2 interaction.

Our observation that independent of the presence of nascent chains 100% of our purified ribosomes can accommodate MAP2, suggests that other factors might be required for its eventual dissociation. Accordingly, such mechanisms could involve subsequent recruitment of ES27L by a different factor, a direct replacement of MAP2, or a combination of both. After NME, NatA would be a plausible successor, as its activity depends on methionine removal. The reported structure of yeast NatE[7] shows that this factor also recruits ES27L. When superimposing the ribosome-bound structures of both factors (Fig. 5), parts of their structures clash, indicating that their binding is mutually exclusive or requires adjustments. Such a factor dependent displacement has recently been shown to occur for bacterial MAP1 in the presence of the PDF enzyme[28] and likely also plays a role in the dissociation of eukaryotic MAP2. Other published structures of RAFs in context of the ribosome, such as NAC[4], RAC[5,6], and SRP[2,3] show a similar scenario, where a concurrent binding would not be possible, unless adaptive structural rearrangements occur at the PTE (Fig. 5).

## MAP2 autoproteolysis and the A-site

Unlike the conserved MAP2 core, the N-terminal extension is highly variable in length and displays species-specific sequence alterations. Overall, these extensions are made up of segmented regions of alternating charged residues. While *Ct*MAP2 only has two charged segments appended to the core domain, the much longer extension found in *Hs*MAP2 harbors four of them. As evidenced by both of our cryo-EM structures, the positively charged patch closest to the core domain serves to recruit ES27L, while the functions of the remaining segments remain elusive.

Analysis of *Ct*MAP2 stability showed a complete degradation of the N-terminus up to the core domain, consistent with the previously reported autoproteolysis of rat MAP2[16] (Supplementary Figs. 15–17). The cleaved mammalian N-terminal fragment plays an important physiological role and protects the initiation factor subunit eIF2α from phosphorylation, a process that centrally controls protein synthesis and gene expression[30]. The observation that *Hs*MAP2 degrades into two stable fragments, implies a functional turnover, where the gradual loss of the N-terminal extension changes the in vivo function of MAP2. The introduction of multiple proteolytic sites within the N-terminus might enable a precise regulation of additional functions of MAP2 beyond its NME activity at the ribosome. The kinetics of this N-terminal remodeling, as well as its functional regulation in vivo remain to be explored.

The structure of the main autoproteolytic product of *Ct*MAP2 in complex with the 80S ribosome revealed that MAP2ΔN is unable to recruit ES27L, PTE binding is attenuated and the protein occupies the A-site within the 60S/40S interface, were it likely has no enzymatic function. In contrast to the highly dynamic binding mode observed at the PTE, A-site binding is tight, resulting in well-defined densities and with contacts formed to essential sites, i.e. the decoding center and the sarcin ricin loop. Strikingly, the conserved insert domain protrudes deepest into the A-site and occupies the space where the incoming

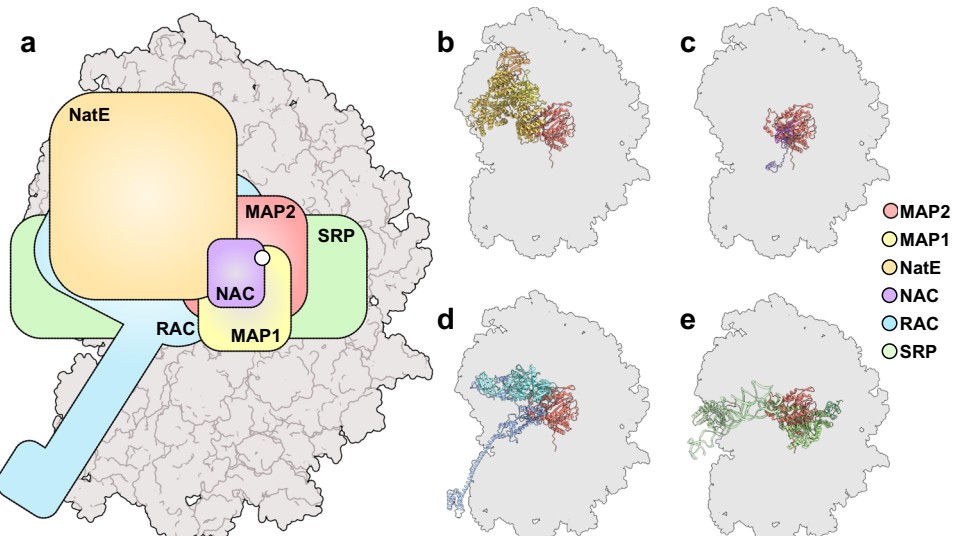

**Fig. 5 | Schematic overview of RAF binding at the eukaryotic tunnel exit.**
**a** Outline of published RAF binding sites on the 80S ribosome, superimposed with the binding site of MAP2. The position of the tunnel exit is indicated by a white circle. The MAP2 binding site overlaps with the binding site of MAP1

(Supplementary Fig. 23), NatE, NAC, RAC, and SRP. Structures of ribosome bound (**b**) NatE[7] 6HD7, (**c**) NAC[4] 7QWR, (**d**) RAC[6] 7Z3N, and (**e**) SRP[4] 7QWQ, superimposed with MAP2.

cognate tRNA is recognized. MAP2ΔN-binding would therefore interfere with the translation cycle and could cause an "elongation arrest", similar to the Alu domain of SRP[31].

The mode of MAP2ΔN-binding to the A-site suggests that MAP2 could already interfere with initiation. In this case, MAP2 would come in close proximity to eIF2, which brings the initiator tRNA into the P-site of the 40S subunit. Such interaction could explain the ability of MAP2 to inhibit eIF2α phosphorylation observed previously[16]. Interestingly, also MAP2-like Ebp1 was reported to bind to the 40S subunit and to inhibit eIF2α phosphorylation[32]. MAP2 might also interfere with translation termination by competing with release factors or with recycling of stalled ribosomes by protein Dom34[33]. Finally, A-site binding to 80S ribosomes might occur only under specific conditions, such as physiological stress or upregulation of MAP2 levels, as reported in different cancer cells[17]. It is important to note, however, that the physiological and pathological roles of this 80S interaction remains to be clarified by more comprehensive and also complementary analyses.

Taken together, our data show that it is the insert domain that specifically directs MAP2 to the site of action on the 80S ribosome, either the PTE or the A-site. Which of these binding sites is realized seems to be impacted mainly by the N-terminus, which can be remodeled by MAP2 itself through proteolytic events at built-in cleavage sites. The segmented architecture of this extension might thereby allow to eject part of the protein sequence to modulate MAP2 function and site of action. Taken together, our cryo-EM structures provide a structural glimpse at the highly dynamic ribosome association of MAP2 and the structural consequences that arise from its proteolytic remodeling, setting the stage for targeted in vivo studies to further elucidate the functional significance of these binding sites.

## Methods
### Sample preparation
Genes encoding *Hs*MAP2 and *Ct*MAP2 were subcloned from *pET24d-His6-linker-MAP2* plasmids into expression vector *pFastBacDuet-HIS10-GSGS-3C-GSGS-MAP2* using in vivo assembly. Clones used in this study are available from the corresponding author upon request. Primers used for the PCR amplification of *map2* genes and the *pFastBacDuet* plasmid can be found in Supplementary Dataset 1. Insect cells (Sf9; Cat. No. 12659017) were grown at 27 °C while shaking at 80 rpm. Cells were

infected at $2 \times 10^6$ cells/ml and expression was carried out over 72 hours. To harvest cells, culture was centrifuged at $1500 \times g$ for 20 min and the pellet was washed once with PBS. Until further use, pellets were frozen in liquid nitrogen and stored at −80 °C. For protein purification by immobilized metal affinity chromatography (IMAC), pellets were resuspended in 20 mM HEPES KOH, 500 mM NaCl, 20 mM Imidazole pH 7.4 with the addition of protease inhibitor (Roche). Cell lysis was carried out with a microfluidizer (Microfluidics Corp.) and lysate was ultracentrifuged for 30 min at $50,000 \times g$ at 4 °C. The cleared lysate was applied to Ni²⁺-IMAC beads and thoroughly washed with lysis buffer. MAPs were cleaved off the column using 3C-protease. Protein was subsequently concentrated in centricons (3 kDa cutoff) and subjected to size exclusion chromatography (SEC). SEC was performed with Superdex 200 16/600 (Cytiva), which was previously equilibrated in SEC buffer (20 mM HEPES KOH, 150 mM KOAc, 5 mM MgOAc₂, 1 mM TCEP pH 7.4). Analytical SEC was performed to validate the purity and monodispersity of all protein samples (Supplementary Fig. 24). Purified protein was frozen in liquid nitrogen and stored at −80 °C until further use.

Non-translating *Hs*80S ribosomes from HeLa cells (HeLa S3; Cat. No. 87110901) were purified as described[21]. Briefly, cells were lysed with detergent and subjected to centrifugation through a sucrose cushion. After puromycin treatment, monosomes were prepared by sucrose gradient centrifugation and isolated ribosomes were concentrated to 1 mg/ml in storage buffer (20 mM HEPES, 100 mM KOAc, 5 MgOAc₂, <3% sucrose, pH 7.4). The preparation of non-translating 80S ribosomes from *Chaetomium thermophilum* (DSMZ-1495) differed by the use of cryo milling for cell lysis. All ribosomes were used directly for cryo-EM grid preparation or stored at −80 °C.

### Cryo-EM grid preparation
Prior to freezing, copper grids were glow-discharged for 60 s in oxygen atmosphere using a Solarus plasma cleaner (Gatan, Inc.). For sample preparation, 200 nM *H. sapiens* ribosomes and 400 nM *C. thermophilum* ribosomes were mixed with *Hs*MAP2 (59 μM final concentration) and *Ct*MAP2/*Ct*MAP2ΔN (66 μM final concentration), respectively. After incubation for 30 min at RT, 3 μl of sample were frozen in liquid ethane on 2/1 Quantifoil Multi A holey carbon supported grids (Quantifoil, Multi A, 400 mesh) using Vitrobot Mark IV (FEI Company). Freezing was performed at 100% humidity, blot force 5,

10 s wait time at 4 °C with Whatman #1 filter papers. Grids were stored in liquid nitrogen until data collection.

## Data collection

For *Hs*MAP2 and *Ct*MAP2ΔN, data was collected on a Titan Krios transmission electron microscope (Thermo Fisher/FEI Company) operated at 300 keV with a K2 Summit direct electron detector (Gatan, Inc.) that collected movies with a pixel size of 1.11 Å/pixel and a magnification of 84,000. The total dose per micrograph was 42.0 and 54.8 e⁻/Å2 for *Hs*MAP2 and *Ct*MAP2ΔN datasets, respectively. Data collection on full-length *Ct*MAP2 with *Ct*80S ribosomes was performed at ESRF[34] on a Titan Krios transmission electron microscope (Thermo Fisher/FEI Company) 300 keV with a K3 Summit direct electron detector (Gatan, Inc.) that collected movies at a pixel size of 0.868 Å/pixel and a magnification of 105,000. The total dose per micrograph was 42.7 e⁻/Å2. EPU was used to set up and monitor the data collection.

## Data processing

Detailed flow-charts describing the processing of *Hs*MAP2, *Ct*MAP2, and *Ct*MAP2ΔN datasets can be found in the supplementary Information. Briefly, movies were imported into cryoSPARC[35] and motion corrected. After patch-CTF-estimation, particles were picked and binned during extraction. Several rounds of 2D-classification were followed by Ab-Initio reconstruction to obtain a preliminary 3D structure. Subsequent rounds of heterogeneous refinement were done to further polish the particle set and to separate 80S ribosomes by their translocation state. Homogenous refinements were performed on the remaining particles and local masks were generated in UCSF Chimera-X[36] to encompass regions of interest. Focused 3D variability analysis was performed with cryoSPARC[25] and 3D-variability display jobs were run to output 20 frame movies as well as 5 or 10 particle clusters for further refinement. Particles separated into 5 or 10 clusters were re-extracted at full size and individually subjected to successive rounds of Ab-initio reconstruction, homogenous refinements and local refinements with masks encompassing ES27L, MAP2, or both with the fulcrum point at the center of mass.

## Cryo-EM model building, refinement, and analysis

As a starting point for model building of the *Hs*MAP2/80S complex, the high-resolution cryo-EM structure of the human ribosome (PDB ID: 6EK0)[37] was rigid body fitted in UCSF Chimera-X[36] into the cryo-EM map, and subsequently also the 1.6 Å crystal structure of *Hs*MAP2 (PDB ID: 1B6A)[38]. For model building of *Ct*MAP2 on the ribosome, the high-resolution cryo-EM structure of the idle *C. thermophilum* ribosome (PDB ID: 7OLC)[26] was rigid body fitted into our cryo-EM map, together with our 1.3 Å crystal structure of *Ct*MAP2. Rigid body fitting was performed in UCSF Chimera-X at full spatial resolution. All further atomic model building was then performed in Coot[39]. Restrained real-space refinement and validation was performed within the PHENIX suite[40,41]. Refinement statistics are shown in Supplementary Table 1. Figures were created with UCSF Chimera-X.

## X-ray structure determination of *Ct*MAP2

*Ct*MAP2 samples were purified as described in the "Sample preparation" section and concentrated to 10 mg/ml in 20 mM HEPES KOH pH 7.4, 150 mM KOAc, 5 mM MgOAc₂ and 1 mM TCEP. Crystallization was performed via the sitting drop vapor diffusion technique at 18 °C. Crystals appeared after 16 h in 0.2 M di-sodium hydrogen phosphate and 20% (w/v) polyethylene glycol 3350 and were cryoprotected by soaking in mother liquor supplemented with 20% (v/v) ethylenglycol. Data were collected at 100 K at beamline ID23-1 at the European Synchrotron Radiation Facility (ESRF, Grenoble). Data were integrated with XDS[42] and scaled using AIMLESS[43] as part of the CCP4 package. The structures were solved by molecular replacement with Phaser-

MR[44] implemented in the PHENIX package[40]. An Alphafold generated structure prediction of *Ct*MAP2 was used as a search model[24]. Coot[39] and phenix.refine[41] of the PHENIX package were used for iterative building and reciprocal refinement cycles. Resulting structural models were validated using Molprobity[45]. Crystallographic data, refinement, and model statistics are summarized in Supplementary Table 2.

## Autoproteolysis of MAP2

To test whether *Ct*MAP2 and *Hs*MAP2 undergo autoproteolysis in vitro, purified protein was concentrated to 1 mg/ml and incubated at 4 °C for 42 days. At different time points ($t_0$, 3 h, 1d, 3d, 4d, 5d, 6d, 7d, 14d, 21d, 28d, 35d and 42d), 5 μl of protein was removed (5 μg) and denatured in Laemmli buffer. Sample was stored at −20 °C and finally subjected to SDS-PAGE and Coomassie staining.

## Mass spectrometry

Coomassie-stained SDS-gel bands of MAP2 (Supplementary Fig. 15) were excised and buffer exchanged for acetonitrile and protein was subsequently reduced by addition of 200 μl of a 10 mM DTT solution in 100 mM ammonium bicarbonate (AmBiC). After incubation at 56 °C for 30 min, proteins were alkylated by the addition of 200 μl of 55 mM chloroacetamide in 100 mM AmBiC and incubated for 20 min in the dark. For acid hydrolysis, samples were incubated for 5 min at room temperature with 50 μl of a 3 M hydrochloric acid solution. The sample was mixed with 700 μl H₂O and microwaved for 10 min. The supernatant was subjected to a reverse phase clean-up step. Peptides were dried and reconstituted in 10 μl of an aqueous solution of 0.1% (v/v) formic acid.

Peptides were analyzed by LC-MS/MS on an Orbitrap Fusion Lumos mass spectrometer (Thermo-Scientific). Peptides were separated using an Ultimate 3000 nano RSLC system (Dionex) equipped with a trapping cartridge (Precolumn C18 PepMap100, 5 mm, 300 μm i.d., 5 μm, 100 Å) and an analytical column (Acclaim PepMap 100. 75 × 50 cm C18, 3 mm, 100 Å) connected to a nanospray-Flex ion source. For detection, hydrolyzed peptides were loaded onto the trap column at 30 μl/min using solvent A (0.1% formic acid) and peptides were eluted using a gradient from 2 to 85% solvent B (0.1% formic acid in acetonitrile) over 30 min at 0.3 μl/min. The Orbitrap Fusion Lumos was operated in positive ion mode with a spray voltage of 2.2 kV and capillary temperature of 275 °C. Full-scan MS spectra with a mass range of 350-1,500 m/z were acquired. The maximum injection time was set to 100 ms. Fragmentation was triggered by HCD with fixed collision energy of 30%. MS2 spectra were acquired with the Ion Trap. The Ion Trap Scan Rate was set to rapid and the maximum injection time to 35 ms.

For data processing, RAW data files from the mass spectrometric analysis were converted into mgf-files using IsobarQuant[46]. The mgf files were used for the Mascot search, which returns a dat-file that was used in IsobarQuant to generate peptide and protein txt-files. These files were finally used for the sequence alignment. The database used for peptide identification can be found as part of supplementary dataset 1. A total of 5 samples were analyzed by Mass Spectrometry without technical replicates or control bands.

## Reporting summary

Further information on research design is available in the Nature Portfolio Reporting Summary linked to this article.

# Data availability

The atomic coordinates have been deposited in the Protein Data Bank under accession codes PDB 8ONX (X-ray structure of *Ct*MAP2ΔN), and for cryo-EM structures 8ONY (*Hs*MAP2 on *Hs*80S ribosomes), 8ONZ (*Ct*MAP2 on *Ct*80S ribosomes), and 8OO0 (*Ct*MAP2ΔN on *Ct*80S ribosomes). The respective cryo-EM volumes have been deposited in the Electron Microscopy Data Bank under accession codes EMD-17002

(*Hs*MAP2), EMD-17003 (*Ct*MAP2), and EMD-17004 (*Ct*MAP2ΔN). For the making of figures and analysis of results, the following published structures were used: MAP2 (1B6A), SRP (7QWQ), RAC (7Z3N), NatE (6HD7), NAC (7QWR), Arx1 (4V8T), Ebp1 (6SXO), MAP1-NAC (8P2K), *Ct*80S (7OLC), and *Hs*80S (6EK0). Mass Spectrometry data are included in the Supplementary information file. Source data are available in the Source Data file.

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

## Acknowledgements

We acknowledge excellent technical support by Marina Pelzl, Britta Klem, Astrid Hendricks, and Silke Adrian, and thank Claudia Siegmann and Jürgen Kopp from the BZH crystallization platform for valuable contributions to the *Ct*MAP2ΔN crystallization. We acknowledge access to the infrastructure of the cryo-EM Network (HDCryoNET) at Heidelberg University, and support by Dirk Flemming (BZH), Lutz Nücker (BZH), and Götz Hofhaus (Bioquant). All cryo-EM grids and preliminary datasets, leading up to the final high-resolution datasets, were screened and acquired in our in-house facilities. We acknowledge the European Synchrotron Radiation Facility for provision of beam time on CM01 at the beginning of the project and we would like to thank Michael Hons for assistance. We also acknowledge access to beamline ID23-1 and thank the ESRF staff for excellent support. We thank the EMBL Proteomics Core Facility and Per Haberkant for collecting MassSpec data on our MAP2 autoproteolysis samples and excellent support. Further, we acknowledge the services SDS@hd and bwHPC supported by the Ministry of Science, Research and Arts Baden-Württemberg and the German Research Foundation through grants INST 35/1314-1 FUGG and INST 35/1134-1 FUGG. This work was supported by the Leibniz Programme of the Deutsche Forschungsgemeinschaft to I.S. (SI 586-6).

## Author contributions

M.A.K., K.W., M.K. and I.S. designed the study and wrote the paper. M.A.K. generated all DNA constructs, purified proteins and ribosomes. Cryo-EM grids were prepared by M.A.K. and data were acquired and processed by M.A.K. and M.K.. M.A.K. and K.W. built the atomic models, and M.A.K., K.W., M.K. and I.S. analyzed the data.

## Funding

## Competing interests

The authors declare no competing interests.
