## [Peer Review File · Nature Communications]

Methionine aminopeptidase 2 and its autoproteolysis product
have different binding sites on the ribosomeReviewer #1 (Remarks to the Author):

As the polypeptide chain emerges from the ribosome, it becomes the substrate for a number of co-translationally interacting chaperones and factors. One of the earliest factors are likely to be the methionine aminopeptidases that catalyze the removal of the initiator methionine from the N-terminus of the nascent polypeptide chain. Although structures of related proteins such as Arx1 and Ebp1 on the ribosome exist and suggest that methionine aminopeptidases (MAP) are also likely to bind analogously, direct visualization of their binding site on the ribosome has so far not been reported. In the study of Klein et al, the authors use cryo-EM to determine structures of eukaryotic methionine aminopeptidase 2 (MAP2) in complex with the eukaryotic ribosome...specifically, human MAP2 on vacant 80S ribosomes and MAP2 from the thermophilic fungi *C. thermophilum* on translational 80S ribosomes, revealing the interaction of the MAP2 with the components of the ribosomal exit site. In addition, the authors make the surprising finding, as eluded to in the title of the manuscript, that an additional binding site on the ribosome is observed for the autoproteolysis product of MAP2 – namely, between the small and large subunit in a site incompatible with continued protein synthesis.

Overall, the manuscript is well-written, nicely illustrated and technical sound. I have only a few comments.

1. The strength of the manuscript is that it provides the first visualization of the MAP2 binding site at the tunnel exit of human and fungal ribosomes. The major weakness is that the authors simply threw purified protein onto isolated 80S ribosomes rather than making the effort to utilize defined functional states. The outcome is that little insight into how the MAP2 protein interacts with the N-terminus of the polypeptide as it emerges from the ribosomal exit site can be gained or how it carries out its function. In the human case, there is no polypeptide chain since the authors released it with puromycin. In the fungal case, the puromycin did not work (for some reason?) and the authors bound the MAP2 to ribosomes that have nascent chains extending out of the tunnel, hence the less defined conformation of MAP2 in this case. It is a shame that the authors did not try to stall ribosomes with defined polypeptide chain lengths to visualize interaction of MAP2 with the nascent chain, however, this is clearly too much to be asked for now in the context of this study.

2. Given the relative modest functional insight from the structures in point 1, the most exciting finding in my opinion is the binding of the autoproteolysis product of MAP2 within the intersubunit interface in a position that is incompatible with translation. Unfortunately, the authors did not take this finding any further, showing that it is physiological relevant and directly showing it can inhibit translation, which would have definitely increased in the value of the study. This is also somewhat surprising given that the authors have the purified protein and could easily monitor whether it does inhibit *in vitro* translation and ascertain at which step. In this regard, it should be pointed out that the complexes were formed with excess protein over ribosomes...on page 24, it states 200nM ribosomes with 59uM HsMAP2 i.e. 300-fold excess, whereas for *C.thermophilum* its written 400mM ribosomes with 66uM – I presume it's a typo and it should be 400nM with 66uM protein i.e. 165-fold excess? The large excesses however do raise the question as to whether the additional binding site is physiological. What are the levels of the MAP2 autoproteolysis product in the cell relative to ribosomes. It would seem relatively easy to monitor the association of the truncated product using sucrose gradients and immunoblotting for example.

3. Lastly, unless I missed it, I did not see any map versus model validations to exclude overfitting. These are pretty standard and should be included in the supplementary information.

Reviewer #2 (Remarks to the Author):

Recommendation:

Publish as it is.

Klein et al. describe several cryo-EM structures of MAP2 from human and the fungus *C. thermophilum* at the 80S ribosome. First, they use a combination of cryo-EM, crystal structure and AlphaFold in order to obtain a high resolution structural basis of MAP2 binding to the 80S ribosome. Next, the authors focus on MAP2 contacts at the ribosomal exit tunnel (PTE) showing that the major determinant for binding to the tunnel exit is the MAP2-specific insert domain and that the second insertion and the N-terminus do not interact with the peptide tunnel exit. Furthermore, the binding of HsMAP2 to idle ribosomes is shown to be rigid whereas the binding of CtMAP2 suggests higher dynamics if nascent chains are present. Structural constraints and possible rotation of MAP2 and the flexible arm ES27L are discussed. Finally, the authors show autoproteolysis for the human and *C. thermophilum* MAP2 which leaves a product missing the N-terminal part of MAP2. This product is expected to lose ES27L binding which is confirmed by the structure but surprisingly the structure does not show changes of MAP2 at the ribosome. Most interestingly, the autocleavage product of MAP2 is strongly stabilized at the A-site suggesting regulatory functions in translation.

Newly synthesized proteins have to be cotranslationally processed at their N-terminus right at the ribosomal exit tunnel in order to become a functional mature protein. Dysfunctional cotranslational regulation leads to protein instability, functional defects and wrong subcellular localizations having severe consequences for cellular proteostasis and dysfunction of N-terminal modifications might lead to human diseases like for example cancer. Several cotranslational biogenesis factors have been identified, like for example the N-terminal methionine excision (MAP1 and MAP2), the eukaryotic nascent polypeptide-associated complex (NAC), N-terminal acetylation (N-terminal acetyl transferases) and targeting to the endoplasmic reticulum (ER) membrane (signal recognition particle, SRP) which use the ribosome as platform to exert their functions. Cotranslational modifications of the nascent protein chains have to be highly coordinated in time, for example N-terminal acetylation requires prior methionine excision, and space (recognition of correct substrates) to ensure functional mature proteins. Although being a fundamental biological mechanism and the fact some of the cotranslational modifications have been already studied for decades, our structural knowledge about binding of these factors to the ribosome and subsequent mechanistics remain still poorly understood.

The data are novel as so far only structures of MAP2 alone without the ribosome were available and provides valuable structural information which will pave the way for further functional and in vivo studies elucidating the complex process of protein maturation. The data are interesting for a broad community working on protein maturation and ribosome-associated factors. In addition, roughly two thirds of all proteins undergo MAP1 or MAP2 processing. Here, structural information about the human MAP2 bound to the ribosome is provided which I consider very important as there might be important differences between species (i.e. yeast has no NAC). The data is limited to structural information and I would like to see more functional data and how the interplay between MAP2, N-terminal acetyltransferase and other factors is coordinated, but I see that this would open a whole new project and delay the publication for a longer period of time. Therefore, I suggest to publish the structural work as it is.

I want to point out that the structural data seem well done, but I am no cryo-EM expert to judge this properly.

Minor question:

Has the autoproteolytic products been analyzed by MS?

Reviewer #3 (Remarks to the Author):

MAP2 is a eukaryotic methionine aminopeptidase, responsible for the co-translational excision of the initiator methionine from the nascent peptide chains. As one of the ribosome-associated factors, it should presumably bind to the peptide tunnel exit (PTE) region of the ribosome to exert its function. Up to date, the interaction details between MAP2 and the ribosome have not been characterized. In this manuscript, Klein et al. determined cryo-EM structures of MAP2 bound to the human and *C. thermophilum* 80S ribosomes. An unexpected and intriguing finding is that MAP2 could also bind to the ribosomal A-site in vitro. This is an interesting observation, which could have

profound biological implications. However, the authors did not make any efforts to exclude the possibility of an *in vitro* artifact. In addition, many conclusions are not directly supported by their experimental data, and some of the figure presentations are not clear. While the determined structures would make a contribution to the field of co-translational nascent chain processing, an extensive revision and supporting experiments would be required ahead of publication.

Major concerns:

(1) The A-site binding could be an *in vitro* artifact. Previous structural studies have reported similar observations for other ribosome-binding factors, such as SecYEG (Mitra et al., 2005). The authors should at least discuss this possibility and try to examine whether the second binding site exists in the cell and is functional relevant (for example, using site-directed crosslinking MS). The authors did not observe the A-site binding of MAP2 in human 80S ribosome, which further questions the physiological relevance of this site.

Related to this concern, both the title and abstract should be revised.

(2) Throughout the manuscript (both figures and text), the authors did not make a distinction between results/data and interpretation/hypothesis. First, in several figures (Figure 1, Figure 3), the authors had a mixed use of cryo-EM model and AF-predicted model without any explanation/annotation in the text or figure legends. From supplementary figures, it appeared that the N-terminal extension of MAP2 was not resolved in the map. But in these figures, all the models have included this extension in the figure panels. Also, by convention, one should show the cryo-EM density maps in one of the figures to reflect the true data.

Second, several conclusions have been made in the abstract. However, none of them is supported by experimental data.

(2.1) In the first paragraph on page 11, the difference on the structural stability of MAP2 relative to the human and *C. thermophilum* ribosomes was attributed to the presence of the nascent chain in the Ct80S ribosome. Based on this observation, the authors made a claim that "The MAP2 interaction is highly robust but can be perturbed by emerging nascent chains, which induce an adaptive tilting of the enzyme". However, no experimental evidence was provided to support this conclusion. The authors should first describe the composition of their 80S-MAP2 structures and clarify whether the two structures contain nascent chains in the tunnel. Second, the interactions between MAP2 and the nascent chain were not present in reported structures. Third, this could also be a species-specific feature rather than a common regulatory mechanism imposed by nascent chains.

(2.2) The next claim in the abstract "In rotation, the MAP2-specific N-terminal extension engages in stabilizing interactions with the long rRNA expansion segment ES27L" (also in the first paragraph on page 7). The interaction between the N-terminal extension of MAP2 and ES27L was derived by aligning the full-length structure of MAP2 predicted by AlphaFold2. This N-terminal extension of MAP2 is unstructured as shown in Figure 1, and without experimental evidence there is no reason to believe that it would be inserted into the major groove of ES27L. The authors could test a few point mutations to see whether the interaction could be weakened.

(2.3) The third claim in the abstract "Loss of this extension by autoproteolytic cleavage impedes the tunnel interaction, while promoting MAP2 to enter the ribosomal A-site".

In paragraph 2 on page 13, "In this dataset, only about 70% of the 80S ribosomes contain CtMAP2ΔN at the PTE (Supplementary Fig. 8)". What are the occupancies of MAP2 in the other two datasets? The 70% occupancy of CtMAP2ΔN at the PTE indicates a high affinity of CtMAP2ΔN to this site of the ribosome. Therefore, it is not reasonable to conclude that "Loss of this extension by autoproteolytic cleavage impedes the tunnel interaction".

In the last paragraph on page 15, "Therefore, it is not obvious, why the full-length protein would not be capable of A-site binding". This sentence is confusing since WT CtMAP2 also binds to the A-

site.

(3) On page 8-9, the descriptions on MAP2-PTE interaction were not well organized. The conserved and not conserved interactions between the species should be clearly stated. For these conserved interactions, the interaction details for both structures could be displayed in a same view. In addition, <1> For the description "In the human case, helix H59 undergoes a large conformational remodeling with the closing loop forming multiple interactions with the insert domain around helix a6", this conformational remodeling lacks figure citation, and the insert domain around helix a6 should be labeled in figure; <2> For "Herein, U2708 forms the main and sole specific contact by piercing into the insert domain where it is read in a Watson-Crick-like manner and stacks in between His391 and Leu403 (Fig. 2A-II)", what does "Watson-Crick-like manner" mean here? And the stacking between U2708 and Leu403 was not shown in Fig. 2A-II. <3> For the description "Interestingly, the binding of CtMAP2 does not trigger a remodeling of helix H59, which remains in the same conformation as in the idle 80S structure, and does not reveal the Watson-Crick read-out", the structural details should be displayed in comparison with that of human one.

Other comments:

(1) Many side-chain interactions were shown in figures. The local EM densities of these protein or RNA segments should be shown to confirm the accuracy of structural models.

(2) In the second paragraph on page 13, "Both CtMAP2 and HsMAP2 showed a complete degradation of the N-terminus up to the core domain, consistent with the previously reported autoproteolysis of rat MAP2 (Supplementary Fig. 7)". How to confirm that the residual segment is the core domain? How to confirm the "complete degradation of the N-terminus"? There is no experimental data.

(3) In the last paragraph on page 6, the description "The central placement allows for a minimal distance of the active site to nascent chain substrates (length < 40 residues) when emerging from the tunnel", should be presented in a figure.

(4) In the last paragraph on page 11, "which touches down onto the phosphate backbone of rRNA helix H24 diametral to Arg361 on the other side of the tunnel exit (Fig. 3-I)". What dose this sentence mean? Please label H24 and Arg361 in figure.

"Measuring the angle of rotation between Arg361 and His316 in the two extreme rotational states of MAP2, a maximum of 15° rotation was determined, which is due to constraints given by the sliding of the MAP2 insert domain over the uL23 surface."; "The second constraint is formed by the protruding 28S rRNA H59. As the rotation is towards the helix, it forms a barrier and MAP2 might finally be sheared-off." All these sentences require supporting figures.

(5) In paragraph 2 on page 14, "Towards the 40S subunit, mostly polar and charged interactions are formed by the rim of the concave side of CtMAP2ΔN to protein eS30 and the extended N-terminus of eS23 (Supplementary Fig. 9)". The "polar and charged interactions" and the "N-terminus of eS23" should be shown or labeled in figures.

REVIEWER COMMENTS

Reviewer #1 (Remarks to the Author):

*As the polypeptide chain emerges from the ribosome, it becomes the substrate for a number of co-translationally interacting chaperones and factors. One of the earliest factors are likely to be the methionine aminopeptidases that catalyze the removal of the initiator methionine from the N-terminus of the nascent polypeptide chain. Although structures of related proteins such as Arx1 and Ebp1 on the ribosome exist and suggest that methionine aminopeptidases (MAP) are also likely to bind analogously, direct visualization of their binding site on the ribosome has so far not been reported. In the study of Klein et al, the authors use cryo-EM to determine structures of eukaryotic methionine aminopeptidase 2 (MAP2) in complex with the eukaryotic ribosome...specifically, human MAP2 on vacant 80S ribosomes and MAP2 from the thermophilic fungi *C. thermophilum* on translational 80S ribosomes, revealing the interaction of the MAP2 with the components of the ribosomal exit site. In addition, the authors make the surprising finding, as eluded to in the title of the manuscript, that an additional binding site on the ribosome is observed for the autoproteolysis product of MAP2 – namely, between the small and large subunit in a site incompatible with continued protein synthesis.*

Overall, the manuscript is well-written, nicely illustrated and technical sound.

We thank the referee for this positive reception of our manuscript.

I have only a few comments.

1. The strength of the manuscript is that it provides the first visualization of the MAP2 binding site at the tunnel exit of human and fungal ribosomes. The major weakness is that the authors simply threw purified protein onto isolated 80S ribosomes rather than making the effort to utilize defined functional states. The outcome is that little insight into how the MAP2 protein interacts with the N-terminus of the polypeptide as it emerges from the ribosomal exit site can be gained or how it carries out its function. In the human case, there is no polypeptide chain since the authors released it with puromycin. In the fungal case, the puromycin did not work (for some reason?) and the authors bound the MAP2 to ribosomes that have nascent chains extending out of the tunnel, hence the less defined conformation of MAP2 in this case. It is a shame that the authors did not try to stall ribosomes with defined polypeptide chain lengths to visualize interaction of MAP2 with the nascent chain, however, this is clearly too much to be asked for now in the context of this study.

We thank the referee for this analysis of possible strengths and weaknesses of our study. We agree that another structure with MAP2 on programmed ribosomes would be nice to have. However, such a dataset will probably not provide detailed information on MAP2 interaction with the N-terminus of a nascent chain. This is based on previously published structures of ribosome associated factors with programmed ribosomes that contain defined nascent chains. However, these were rarely resolved beyond the peptide tunnel exit, and there is not a single case where a nascent chain was continuously traced up to the ribosome associated factor (e.g., **NAC** (pdb: 7QWR), **NatB** (pdb: 8bip), **NatE** (pdb: 6HD7), **SRP** (pdb: 7QWQ)). The mix of random nascent chains present in our *Chaetomium* ribosomes likely contributed to the rotation of MAP2 at the PTE. Programmed ribosomes with a single nascent chain might not have revealed this intriguing mode of binding.

*2. Given the relative modest functional insight from the structures in point 1, the most exciting finding in my opinion is the binding of the autoproteolysis product of MAP2 within the intersubunit interface in a position that is incompatible with translation. Unfortunately, the authors did not take this finding any further, showing that it is physiological relevant and directly showing it can inhibit translation, which would have definitely increased in the value of the study. This is also somewhat surprising given that the authors have the purified protein and could easily monitor whether it does inhibit in vitro translation and ascertain at which step. In this regard, it should be pointed out that the complexes were formed with excess protein over ribosomes...on page 24, it states 200nM ribosomes with 59uM HsMAP2 i.e. 300-fold excess, whereas for *C.thermophilum* its written 400mM ribosomes with 66uM – I presume it's a typo and it should be 400nM with 66uM protein i.e. 165-fold excess? The large excesses however do raise the question as to whether the additional binding site is physiological. What are the levels of the MAP2 autoproteolysis product in the cell relative to ribosomes. It would*

seem relatively easy to monitor the association of the truncated product using sucrose gradients and immunoblotting for example.

We thank the referee for the suggestions of future experiments to further elucidate the function of the autoproteolysis product. Unfortunately, for *C. thermophilum* an *in vitro* translation system is not available to test our purified CtMAP2 Δ N for stalling.

For the excess of MAP2: indeed, we used a 165-fold excess of MAP2 in the *Chaetomium* dataset. We thank the referee for pointing out this typing error and have corrected it in the manuscript.

The literature on MAP2 suggests strongly varying expression levels in different tissues, under different growth conditions, in different disease models or different cell cycles and stress conditions. We agree that analysis of MAP2 autoproteolysis levels will be an interesting and valuable experiment. However, for the current manuscript we chose to focus on the structural aspects. These data however allow to design more targeted experiments to explore the physiological significance of the two binding sites in the future.

3. Lastly, unless I missed it, I did not see any map versus model validations to exclude overfitting. These are pretty standard and should be included in the supplementary information.

The data processing and validation statistics provided in the supplementary material are in accordance with current standards as seen also in recently published structures of ribosome associated factors at the ribosome (NatB: <https://doi.org/10.1371/journal.pbio.3001995>, NAC-SRP: <https://doi.org/10.1126/science.abl6459>, NAC-MAP: <https://doi.org/10.1126/science.adg3297>).

The validated fit-to-map metrics we give in Table S1 is the respective Phenix CC(mask) parameter. We now added the calculated dFSC (0.143) map-to-model (mask) resolution.

Reviewer #2 (Remarks to the Author):

*Recommendation:
Publish as it is.*

We thank the referee for this positive statement and clear recommendation.

Klein et al. describe several cryo-EM structures of MAP2 from human and the fungus C. thermophilum at the 80S ribosome. First, they use a combination of cryo-EM, crystal structure and AlphaFold in order to obtain a high resolution structural basis of MAP2 binding to the 80S ribosome. Next, the authors focus on MAP2 contacts at the ribosomal exit tunnel (PTE) showing that the major determinant for binding to the tunnel exit is the MAP2-specific insert domain and that the second insertion and the N-terminus do not interact with the peptide tunnel exit. Furthermore, the binding of HsMAP2 to idle ribosomes is shown to be rigid whereas the binding of CtMAP2 suggests higher dynamics if nascent chains are present. Structural constraints and possible rotation of MAP2 and the flexible arm ES27L are discussed. Finally, the authors show autoproteolysis for the human and C. thermophilum MAP2 which leaves a product missing the N-terminal part of MAP2. This product is expected to lose ES27L binding which is confirmed by the structure but surprisingly the structure does not show changes of MAP2 at the ribosome. Most interestingly, the autocleavage product of MAP2 is strongly stabilized at the A-site suggesting regulatory functions in translation.

Newly synthesized proteins have to be cotranslationally processed at their N-terminus right at the ribosomal exit tunnel in order to become a functional mature protein. Dysfunctional cotranslational regulation leads to protein instability, functional defects and wrong subcellular localizations having severe consequences for cellular proteostasis and dysfunction of N-terminal modifications might lead to human diseases like for example cancer. Several cotranslational biogenesis factors have been identified, like for example the N-terminal methionine excision (MAP1 and MAP2), the eukaryotic nascent polypeptide-associated complex (NAC), N-terminal acetylation (N-terminal acetyl transferases) and targeting to the endoplasmic reticulum (ER) membrane (signal recognition particle,

SRP) which use the ribosome as platform to exert their functions. Cotranslational modifications of the nascent protein chains have to be highly coordinated in time, for example N-terminal acetylation requires prior methionine excision, and space (recognition of correct substrates) to ensure functional mature proteins. Although being a fundamental biological mechanism and the fact some of the cotranslational modifications have been already studied for decades, our structural knowledge about binding of these factors to the ribosome and subsequent mechanistic remain still poorly understood.

The data are novel as so far only structures of MAP2 alone without the ribosome were available and provides valuable structural information which will pave the way for further functional and in vivo studies elucidating the complex process of protein maturation. The data are interesting for a broad community working on protein maturation and ribosome-associated factors. In addition, roughly two thirds of all proteins undergo MAP1 or MAP2 processing. Here, structural information about the human MAP2 bound to the ribosome is provided which I consider very important as there might be important differences between species (i.e. yeast has no NAC). The data is limited to structural information and I would like to see more functional data and how the interplay between MAP2, N-terminal acetyltransferase and other factors is coordinated, but I see that this would open a whole new project and delay the publication for a longer period of time. Therefore, I suggest to publish the structural work as it is.

We are very happy about this positive reception and thorough analysis of our study.

I want to point out that the structural data seem well done, but I am no cryo-EM expert to judge this properly.

Minor question:

Has the autoprolytic products been analyzed by MS?

Yes, the autoprolysis product has been analyzed by MS. We have added the results of the MS analysis to the manuscript (**Supplementary Figs. 13 and 14**).

Reviewer #3 (Remarks to the Author):

MAP2 is a eukaryotic methionine aminopeptidase, responsible for the co-translational excision of the initiator methionine from the nascent peptide chains. As one of the ribosome-associated factors, it should presumably bind to the peptide tunnel exit (PTE) region of the ribosome to exert its function. Up to date, the interaction details between MAP2 and the ribosome have not been characterized. In this manuscript, Klein et al. determined cryo-EM structures of MAP2 bound to the human and C. thermophilum 80S ribosomes. An unexpected and intriguing finding is that MAP2 could also bind to the ribosomal A-site in vitro. This is an interesting observation, which could have profound biological implications. However, the authors did not make any efforts to exclude the possibility of an in vitro artifact. In addition, many conclusions are not directly supported by their experimental data, and some of the figure presentations are not clear. While the determined structures would make a contribution to the field of co-translational nascent chain processing, an extensive revision and supporting experiments would be required ahead of publication.

We are thankful for the in-depth analysis of our manuscript, and for the helpful and constructive criticism.

The implications of MAP2 function in disease has sparked decades of research towards developing specific drugs that target this metalloprotease. All of this work has been done without knowing how MAP2 interacts with the eukaryotic ribosome. In our study, we not only close this gap, but also reveal the striking consequences that autoprolysis has on the 80S interaction. In our opinion these structures are vital for the continued exploration of MAP2 as a potential drug target, and important for understanding the intricate function that MAP2 assumes inside the cell.

Our work succeeded in elucidating the molecular details of the MAP2-80S interaction. We strongly feel that the work that we present is concise and coherent. We are however grateful for the detailed recommendations, which we incorporated in the manuscript.

Below, we have addressed each point separately and hope that the revised manuscript and our replies alleviate any concerns and possible misconceptions.

Major concerns:

(1) The A-site binding could be an in vitro artifact. Previous structural studies have reported similar observations for other ribosome-binding factors, such as SecYEG (Mitra et al., 2005). The authors should at least discuss this possibility and try to examine whether the second binding site exists in the cell and is functional relevant (for example, using site-directed crosslinking MS). The authors did not observe the A-site binding of MAP2 in human 80S ribosome, which further questions the physiological relevance of this site.

Related to this concern, both the title and abstract should be revised.

Since we obtained our sample by saturating ribosomes *in vitro*, we agree that the observation of an unexpected binding site should be thoroughly discussed. We too questioned the functional relevance of the A-site interaction, when we first obtained the structure. Observations of other ribosome associated factors (RAFTs), such as SecYEG at non-physiological binding sites made us acutely aware of this. For the following reasons, we became however strongly convinced that the A-site interactions is indeed physiologically relevant, and not an artifact introduced by our experimental design:

- 1) Unlike previous observations of RAFTs at non-physiological binding sites, the MAP2 interaction is resolved at high resolution, highlighting the rigidity of the contacts formed by MAP2 in the A-site. Most contacts are formed by conserved structural elements that are unique to MAP2, e.g., R361 and the N-terminal helices. R361 coincidentally also assumes a major role at the PTE by acting as the pivot point of rotation. Interestingly, the N-terminal helices serve no active role in PTE binding and a function could thus far not be assigned.
- 2) MAP2 interacts with some of the most conserved parts of the ribosome in the A-site. R361 interacts with the decoding center and induces conformational changes, and the flexible N-terminal helices become rigid as they engage in pi-stacking interactions with the sarcin-ricin-loop. In our opinion, the observation that not any, but conserved and functionally important features of the A-site interact with MAP2 and are conformationally remodeled, are strongly in favor of a specific interaction and physiological relevance.
- 3) A-site binding appears to be dependent of prior autoproteolysis and specifically binds ribosomes in the non-rotated state. Our dataset of the C α MAP2 autoproteolysis product revealed that MAP2 Δ N had bound the A-site with similar occupancy as eEF2 (around 25% of all 80S). In contrast, full length C α MAP2 only bound the A-site of ~5% of all 80S particles, even though the sample was prepared the same way as for C α MAP2 Δ N. Our *in vitro* autoproteolysis experiment showed that full length C α MAP2 naturally degrades to its autoproteolysis product at 1 mg/ml at 4°C. The cryoEM sample of full length C α MAP2 was prepared at much higher concentrations and at room temperature. It is likely that a subset of MAP2 had auto-proteolyzed before the sample was frozen, and that this subset bound the A-site, and not the full-length protein. Even if the full-length protein would be able to enter the A-site, it clearly does so at lower affinity, as the full-length protein showed a 5-fold decrease in A-site occupancy.
- 4) Furthermore, we did not observe A-site binding for full length HsMAP2. In contrast to *Chaetomium* ribosomes, the human 80S monosomes are almost exclusively in the rotated state and bound to eEF2. Even if a subset of HsMAP2 auto-proteolyzed, we wouldn't be able observe A-site binding in the absence of ribosomes in the non-rotated state.

(2) Throughout the manuscript (both figures and text), the authors did not make a distinction between results/data and interpretation/hypothesis.

We respectfully disagree with the generic statement that we do not make a distinction between data and interpretation. Nonetheless, we are happy to address the individual concerns and changed the text and figures to avoid possible confusion.

First, in several figures (Figure 1, Figure 3), the authors had a mixed use of cryo-EM model and AF-predicted model without any explanation/annotation in the text or figure legends. From supplementary figures, it appeared that the N-terminal extension of MAP2 was not resolved in the map. But in these figures, all the models have included this extension in the figure panels. Also, by convention, one should show the cryo-EM density maps in one of the figures to reflect the true data.

We only use an AF-predicted model in one case – for the largely unstructured MAP2 N-terminus. We acknowledge that this partially predicted element shown in figure 1 could be confusing and have removed the N-terminal extension from all main text figures to only show results from the cryoEM data. The composite model has been moved to the supplements and the predicted N-terminus has been highlighted (**Supplementary Fig. 5**).

The cryoEM density is shown in the supplement, and in addition we have added several new supplementary figures.

Second, several conclusions have been made in the abstract. However, none of them is supported by experimental data.

We respectfully disagree with this generic statement. We do not make conclusions without data, but we have revised the phrasing of our abstract to avoid such misconception. Below, we have addressed each concern in detail and made our best effort to implement the referees suggestions.

(2.1) In the first paragraph on page 11, the difference on the structural stability of MAP2 relative to the human and C. thermophilum ribosomes was attributed to the presence of the nascent chain in the Ct80S ribosome. Based on this observation, the authors made a claim that “The MAP2 interaction is highly robust but can be perturbed by emerging nascent chains, which induce an adaptive tilting of the enzyme”. However, no experimental evidence was provided to support this conclusion. The authors should first describe the composition of their 80S-MAP2 structures and clarify whether the two structures contain nascent chains in the tunnel.

In our cryoEM map of Ct80S ribosomes, a mix of random nascent chains occupies the exit tunnel, as detailed in a previous publication (Kišonaitė et al. 2022). Human 80S monosomes do not show such density, indicating the absence of nascent chains. To clarify, we have added a figure to the supplementary information, showing the presence of nascent chains only in the *Chaetomium* ribosomes (**Supplementary Fig. 11**).

Second, the interactions between MAP2 and the nascent chain were not present in reported structures. Third, this could also be a species-specific feature rather than a common regulatory mechanism imposed by nascent chains.

An interaction between CtMAP2 and nascent chains is not visible in our dataset. However, this is to be expected, as our sample does not contain programmed ribosomes. The ribosomes that were isolated from *Chaetomium* cells likely contain a mixture of nascent chains of different length and sequence. Of note, in previous published structures of programmed ribosomes containing a defined nascent chain and the corresponding ribosome associated factor, the nascent chain is rarely resolved beyond the tunnel exit, and to date continuous density up to the ribosome associated factor has not been observed (e.g., **NAC** (pdb: 7QWR), **NatB** (pdb: 8bip), **NatE** (pdb: 6HD7), **SRP** (pdb: 7QWQ).

Regarding the rotation of CtMAP2, we agree that we cannot rule out a species-specific feature, and we are happy to more clearly highlight this possibility in the results section. The text has been changed accordingly. Based on the structural data however, we find it more plausible that nascent chains would prevent a stable association of CtMAP2 at the PTE. MAP2 sits centrally on the PTE and would leave little to no room for an emerging nascent chain. The backwards rotation offers an explanation as to why 100% of all ribosomes can accommodate CtMAP2 at the tunnel exit despite the

mix of random nascent chains.

(2.2) The next claim in the abstract “In rotation, the MAP2-specific N-terminal extension engages in stabilizing interactions with the long rRNA expansion segment ES27L” (also in the first paragraph on page 7). The interaction between the N-terminal extension of MAP2 and ES27L was derived by aligning the full-length structure of MAP2 predicted by AlphaFold2. This N-terminal extension of MAP2 is unstructured as shown in Figure 1, and without experimental evidence there is no reason to believe that it would be inserted into the major groove of ES27L. The authors could test a few point mutations to see whether the interaction could be weakened.

The positions of MAP2 and the expansion segment are well resolved in our cryoEM map. However, the N-terminal region of MAP2 that bridges the short gap of a few angstroms to the expansion segment is visible but not well resolved, due to its intrinsic flexibility.

The CtMAP2 N-terminal extension is comprised of a negatively charged stretch at the very N-terminus, followed by a positively charged region directly adjacent to the core domain. From the physico-chemical properties and supported by our structural data, it is clear that the positively charged region recruits the negatively charged rRNA. Our dataset of the CtMAP2 autoproteolysis product showed a loss of the ES27L interaction, further supporting the role of the N-terminal extension.

AlphaFold predictions of the N-terminal extension allow to place the positively charged patch in the weak EM density and through the major groove of ES27L. We find this plausible as the widened major groove next to the closing tetraloop could perfectly accommodate the MAP2 extension as often found in RNA recognition (for interaction of arginine rich motifs (ARMs) with RNA see e.g. Grotwinkel et al. 2014; Kišonaitė et al. 2023).

Point mutations in the positively charged region would likely weaken this interaction. However, we have already analyzed the deletion of the N-terminus by cryoEM and reported its significance. In our opinion, producing mutants in insect cells and establishing a method to measure and compare binding affinities is interesting, but beyond the scope of this story. In addition, such an experiment would not clarify whether or not the extension passes through the major groove, but only evaluate the contribution of individual residues to ribosome binding.

(2.3) The third claim in the abstract “Loss of this extension by autoproteolytic cleavage impedes the tunnel interaction, while promoting MAP2 to enter the ribosomal A-site”.

In paragraph 2 on page 13, “In this dataset, only about 70% of the 80S ribosomes contain CtMAP2 Δ N at the PTE (Supplementary Fig. 8)”. What are the occupancies of MAP2 in the other two datasets? The 70% occupancy of CtMAP2 Δ N at the PTE indicates a high affinity of CtMAP2 Δ N to this site of the ribosome. Therefore, it is not reasonable to conclude that “Loss of this extension by autoproteolytic cleavage impedes the tunnel interaction”.

The occupancies of MAP2 for the other datasets are shown in supplementary figures 1 and 2. Since the sample contained a 165-fold molar excess of MAP2, an occupancy of 70% does not necessarily suggest a high affinity for the PTE. For the full-length proteins (*Hs*MAP2 and CtMAP2), 100% of all 80S ribosomes were in complex with MAP2. The autoproteolysis product of CtMAP2 could only bind 70% of all ribosomes, even though the sample was prepared in the same way. In comparison to the full-length protein, the PTE interaction is therefore clearly attenuated.

In the last paragraph on page 15, “Therefore, it is not obvious, why the full-length protein would not be capable of A-site binding”. This sentence is confusing since WT CtMAP2 also binds to the A-site.

We do find a very small amount of CtMAP2 in the A-site. As discussed above, we strongly believe that this small fraction of particles had undergone autoproteolysis and that the full-length protein is either incapable of entering this binding site, or can only bind the A-site with low affinity.

(3) On page 8-9, the descriptions on MAP2-PTE interaction were not well organized. The conserved and not conserved interactions between the species should be clearly stated. For these conserved interactions, the interaction details for both structures could be displayed in a same view.

In addition, <1> For the description “In the human case, helix H59 undergoes a large conformational remodeling with the closing loop forming multiple interactions with the insert domain around helix a6”, this conformational remodeling lacks figure citation, and the insert domain around helix a6 should be

labeled in figure; <2> For “Herein, U2708 forms the main and sole specific contact by piercing into the insert domain where it is read in a Watson-Crick-like manner and stacks in between His391 and Leu403 (Fig. 2A-II)”, what does “Watson-Crick-like manner” mean here? And the stacking between U2708 and Leu403 was not shown in Fig. 2A-II. <3> For the description “Interestingly, the binding of CtMAP2 does not trigger a remodeling of helix H59, which remains in the same conformation as in the idle 80S structure, and does not reveal the Watson-Crick read-out”, the structural details should be displayed in comparison with that of human one.

We apologize for the mismatch of figure detail and text. We are thankful for the helpful recommendations and made an effort to improve the figures and their description, and added additional details and supplementary figures.

Other comments:

(1) Many side-chain interactions were shown in figures. The local EM densities of these protein or RNA segments should be shown to confirm the accuracy of structural models.

To show the accuracy of our model and to highlight important interactions we created several additional supplementary figures (**Supplementary Figs. 6, 9, 12, 16 and 18**).

(2) In the second paragraph on page 13, “Both CtMAP2 and HsMAP2 showed a complete degradation of the N-terminus up to the core domain, consistent with the previously reported autoproteolysis of rat MAP2 (Supplementary Fig. 7)”. How to confirm that the residual segment is the core domain? How to confirm the “complete degradation of the N-terminus”? There is no experimental data.

We have added MassSpec data to the manuscript, which confirm that it is indeed the N-terminus that is degraded (**Supplementary Fig. 14**).

In addition, the size of the emerging autoproteolysis product is identical to that of the cloned CtMAP2 Δ N construct on an SDS-gel (**Supplementary Fig. 20**).

(3) In the last paragraph on page 6, the description “The central placement allows for a minimal distance of the active site to nascent chain substrates (length < 40 residues) when emerging from the tunnel”, should be presented in a figure.

We do not think that a figure is necessary to show the distance of the MAP2 active site to the PTE, since we will deposit all atomic models to the pdb upon publication. The length of nascent chains was calculated by assuming an average C α distance of 3.9 Å per amino acid.

(4) In the last paragraph on page 11, “which touches down onto the phosphate backbone of rRNA helix H24 diametral to Arg361 on the other side of the tunnel exit (Fig. 3-I)”. What dose this sentence mean? Please label H24 and Arg361 in figure.

“Measuring the angle of rotation between Arg361 and His316 in the two extreme rotational states of MAP2, a maximum of 15° rotation was determined, which is due to constraints given by the sliding of the MAP2 insert domain over the uL23 surface.”; “The second constraint is formed by the protruding 28S rRNA H59. As the rotation is towards the helix, it forms a barrier and MAP2 might finally be sheared-off.” All these sentences require supporting figures.

We agree and have rephrased the sentences in the last paragraph of page 11 and created additional figures for the supplements (**Supplementary Fig. 12**).

(5) In paragraph 2 on page 14, “Towards the 40S subunit, mostly polar and charged interactions are formed by the rim of the concave side of CtMAP2 Δ N to protein eS30 and the extended N-terminus of eS23 (Supplementary Fig. 9)”. The “polar and charged interactions” and the “N-terminus of eS23” should be shown or labeled in figures.

We included a supporting figure in the supplementary information (**Supplementary Fig. 16**).

Reviewer #2 (Remarks to the Author):

Recommendation:
Publish as it is.

I refer to my comments in the first revision where I was already in favor of publishing it as the structural work it is which will pave the way for further functional and in vivo studies elucidating the complex process of protein maturation. My minor question if the fragment has been analyzed my MS has been answered. In response to other reviewers' comments the manuscript has been amended and supplementary figures added.

Reviewer #3 (Remarks to the Author):

The authors have addressed my concerns.

REVIEWERS' COMMENTS

Reviewer #2 (Remarks to the Author):

*Recommendation:
Publish as it is.*

We thank the referee for this clear and positive statement.

I refer to my comments in the first revision where I was already in favor of publishing it as the structural work it is which will pave the way for further functional and in vivo studies elucidating the complex process of protein maturation. My minor question if the fragment has been analyzed my MS has been answered. In response to other reviewers' comments the manuscript has been amended and supplementary figures added.

We are grateful that the referee took the time to read and comment on our manuscript in detail and in a very constructive manner.

Reviewer #3 (Remarks to the Author):

The authors have addressed my concerns.

We are thank the referee for this clear and positive statement.